REGISTERED REPORT PROTOCOL

# Telehealth in outpatient care for children and adolescents with chronic conditions during the COVID-19 pandemic: A scoping review protocol

**Larissa Karoline Dias da Silva Casemiro**[1‡], **Luís Carlos Lopes-Júnior**[2‡], **Fabrine Aguilar Jardim**[1], **Mariane Caetano Sulino**[1], **Regina Aparecida Garcia de Lima**[1] *

**1** University of São Paulo at Ribeirão Preto College of Nursing, Ribeirão Preto, SP, Brazil, **2** Health Sciences Center at the Federal University of Espírito Santo (UFES), Vitoria, ES, Brazil

‡ LKDSC and LCLJ equally contributed and joint first authors on this work.
* limare@eerp.usp.br

This is a Registered Report and may have an associated publication; please check the article page on the journal site for any related articles.

## Abstract

### Introduction

Outpatient care for children and adolescents with chronic conditions needs to be continuous and programmed, encompassing comprehensive care, with periodically scheduled consultations, exams, and procedures, to promote quality of life and reduce mortality. In the context of the new coronavirus pandemic, however, outpatient care for children and adolescents with chronic conditions, in person, was hampered in favor of social isolation, a necessary sanitary measure to reduce and prevent the spread of Coronavirus Disease 2019. In response to this need, studies suggest telehealth in pediatrics as a fertile and expanding field especially in times of pandemics. Here, we aimed to map the evidence related to telehealth in outpatient care for children and adolescents with chronic conditions during the COVID-19 pandemic, to identify which strategies were implemented and their impacts on the continuity of care.

### Methods

A scoping review protocol is reported and guided by the Scoping Reviews Manual of the Joanna Briggs Institute. The search for evidence will cover the following databases: MEDLINE/PubMed, Cochrane Libary; Embase; Web of Science; Scopus; Cinahl and PsycINFO, plus additional sources, such as The British Library, Google Scholar, and Preprints [medRXiv]. No date or language restrictions will be employed in this scoping review. Two independent researchers will conduct the search strategy, study selection, data charting, and data synthesis.

### Results

The findings will be presented through tables, charts, narrative summaries, and assessed based on the type of data charted as well as outcomes. Additionally, the meaning of these findings will be considered as they relate to the guiding question, the characterization and

**Data Availability Statement:** Lopes-Júnior LC, Casemiro LKDS, Sulino MC, Jardim FA, Lima RAG. Telehealth in outpatient care for children and adolescents with chronic conditions during the COVID-19 pandemic: a scoping review protocol. 20121. Open Science Framework Repository. https://doi.org/10.17605/OSF.IO/EKF4R.

**Funding:** This study protocol is supported by the Coordination for the Improvement of Higher Education Personnel (CAPES), Notice: Emergency Action/COVID-19. Process Number: 88887.508473/2020-00. Doctoral scholarship to the first author (LKDSC).

**Competing interests:** The authors have declared that no competing interests exist.

measurement of the impact of different telehealth modalities used in outpatient care for children and adolescents with chronic conditions during the COVID-19 pandemic, and the implications for practice and further research.

## Discussion

To the best of our knowledge, this will be the first scoping review to look specifically at the telehealth modalities to be used in outpatient care for children and adolescents with chronic conditions during the COVID-19 pandemic. We expect that our results will be of interest to practitioners as well as researchers concerned with this particular emerging issue in the pandemic context. Also, the plans for the dissemination of this study comprise peer-reviewed publication and conference presentations.

## Trial registration

**Open Science Framework Registration:** osf.io/5pqgu.

## Introduction

Chronic conditions encompass a broad category of health problems with biological, psychological, or cognitive causes [1]. These include transmissible diseases such as HIV/AIDS and tuberculosis; non-communicable diseases such as cardiovascular disease, cancer, chronic respiratory diseases, diabetes; long-term mental disorders, and structural disabilities [2].

The prevalence of chronic conditions in childhood has increased by more than 400% in the last 50 years, favoring the occurrence of comorbidities, a large number of hospitalizations, and a high degree of mortality [3]. A retrospective observational study in a pediatric intensive care unit of a hospital in London found that 46% of hospitalized patients had at least one chronic health condition. Also, these children remained hospitalized longer, increasing the chances of mortality [4]. A prospective, multicenter, observational cohort study conducted in 19 pediatric care units of hospitals in Argentina identified that 48% of hospitalized children had one or more chronic conditions, the most prevalent being cardiovascular diseases (22.24%), neuromuscular diseases (18.75%), and genetic diseases (17.7%) [5].

Children and adolescents with chronic conditions demand high hospital readmission rates, as exemplified in a cohort study in 37 pediatric hospitals in the USA with a sample of 69,294 children. Of the total, 18.8% were readmitted to the service, 89.0% of whom had complex chronic conditions, 52.6% used technological devices, and 2.9% were readmitted more than four times within a year [6].

In this sense, outpatient care for children and adolescents with chronic conditions needs to be continuous and programmed, encompassing comprehensive care, with periodically scheduled consultations, exams, and procedures [1], to promote quality of life and reduce mortality. In the context of the new coronavirus pandemic, however, outpatient care for children and adolescents with chronic conditions, in person, was hampered [7] in favor of social isolation, a necessary sanitary measure to reduce and prevent the spread of Coronavirus Disease 2019 (COVID-19) [8].

Research has shown that COVID-19 can lead to more severe complications in people with chronic conditions [9, 10], which reinforces the recommendations of social isolation for this audience. Thus, it is necessary to reflect on innovative outpatient care models that contribute to the continuity of care and comprehensive health care for patients with chronic conditions.

In response to this need, studies suggest telehealth in pediatrics as a fertile and expanding field, especially in times of a pandemic [11–13]. According to the World Health Organization, telehealth refers to health services at a distance, synchronously or asynchronously, through information and communication technologies. This modality makes it possible to carry out clinical care, diagnoses, treatments, prevention of diseases and injuries, research, and evaluation. Another potential use is in continuing education for health professionals [14].

Telehealth is being used in several countries worldwide [15–21], especially in non-emergency/routine care and in cases where services do not require direct patient-provider interaction such as providing psychological services, mainly in the current scenario of COVID-19 pandemic in order to nullify the barriers imposed by the COVID-19 pandemic. Additionally, telehealth can become a basic need for the general population, health care providers, and patients with COVID-19, mainly when people are in quarantine, enabling patients in real time through contact with health care provider for advice on their health problems.

Telehealth can positively affect the care of children and adolescents with chronic conditions. They need long-term, programmed care, but traveling to the health service can lead to a greater risk of contagion. Therefore, innovative care models are needed, such as care provision using remote methods. Hence, this study aims to map the evidence related to telehealth in outpatient care for children and adolescents with chronic conditions during the COVID-19 pandemic. Another aim is to identify the strategies implemented and their impacts on the continuity of care.

## Methods

This study is a scoping review (ScR) type, which aims to map the key concepts of a particular field of research to clarify definitions and conceptual limits. Additional aims are to identify evidence, analyze knowledge gaps and examine how research is conducted in a given field [22], providing a descriptive analysis of the included studies [23].

The methodological framework described by Arksey and O'Malley [22], and further outlined in the Scoping Reviews Manual [24] of the Joanna Briggs Institute (JBI) serves as a guide for this ScR. The five steps of the ScR include (I) identifying the research question; (II) identifying the relevant studies; (III) selecting the studies; (IV) charting the data; and (V) collating, summarizing, and reporting the results; and will be depicted as follows. Additionally, this protocol has been registered within the Open Science Framework (osf.io/5pqgu). This ScR will be reported following the Preferred Reporting Items for Systematic Review and Meta-Analyses extension for Scoping Reviews (PRISMA-ScR) [25].

### Step 1: Identifying the research question

For the construction of the research question, the PCC (Population, Concept, and Context) framework was applied to identify the main concepts and is described in detail as follows:

### Population

The target population of this scoping review will be children and adolescents aged 0–19 years with chronic conditions. The World Health Organization (WHO) defines a child as a person under 19 years of age [26].

WHO defines chronic conditions as non-communicable diseases (e.g., diabetes, heart disease, lung diseases, and cancer), persistent communicable diseases (e.g., HIV/AIDS and tuberculosis), long-term mental disorders (e.g., schizophrenia and depression), and ongoing physical impairments or structural problems (e.g., amputations, blindness, and musculoskeletal disorders) [2].

## Concept

This scoping review will include studies that address telehealth in child and adolescent care with chronic conditions during the COVID-19 pandemic due to the interruption of face-to-face care in outpatient clinics. For this study, telehealth, according to WHO, refers to the use of information and communication technologies for remote diagnosis and treatment of patients, aiming to promote individual and family health [14].

## Context

The context analyzed in this scoping review is outpatient care, which refers to any healthcare consultation, procedure, treatment, or other service that is administered without an overnight stay in a hospital or medical facility. Routine physical examinations with a primary care provider are one common type of outpatient care, but the outpatient market has grown to include services such as: diagnostic imaging like x-rays and MRIs; bloodwork and urine tests; physical therapy; chemotherapy and radiation treatments etc. Additionally, certain surgical procedures like hip and knee replacements, dental surgery, gastric bypass, breast augmentation, and others can even be performed in the outpatient setting. These services are administered in a variety of different outpatient facilities. Some examples include primary care clinics, community health centers, urgent care clinics, ambulatory surgery centers, and even some pharmacies for basic healthcare consultations and immunizations [27]. In this scoping review, we link outpatient care to COVID-19—a disease caused by a new type of beta-coronavirus, capable of causing respiratory, enteric, liver, and neurological conditions [28]. COVID-19 was first identified in a group of patients with pneumonia in December 2019 in Wuhan, China [28], and WHO declared it a pandemic on March 11, 2020 [29].

Thus, after applying the PCC framework [22], the research question raised was: "What are the scientific productions available in the literature on the use of telehealth in outpatient care for children and adolescents with chronic conditions during the COVID-19 pandemic?"

## Step 2: Identifying the relevant studies

The search for evidence will cover the following databases: Medical Literature Analysis and Retrieval System Online (MEDLINE) via PubMed, Cochrane Libary; Excerpta Medica database (Embase); Web of Science; Scopus; Cumulative Index to Nursing and Allied (Cinahl) and Psychology Information (PsycINFO), plus additional sources, such as The British Library (UK), Google Scholar, and Preprints [medRXiv]. For the search strategy, we will include a combination of subject headings, e.g., Medical Subject Headings (MeSH), Emtree, Cinahl headings, Thesaurus, [30, 31]. Additionally, the reference list of all included studies will be reviewed for additional relevant studies. No date or language restrictions will be employed in this scoping review.

To structure the search, as already mentioned, searches were carried out in MEDLINE/ PubMed to find the best combination of MeSH terms, synonyms as well as free text words, which will be later adjusted for each electronic database. Following the PCC framework (Population/Concept/Context), a pilot search strategy was established for MEDLINE/PubMed (Table 1):

## Step 3: Selecting the studies

- *Inclusion criteria*: all the primary studies, experience reports, dissertations, and theses related to the use of telehealth in outpatient care for children and adolescents with chronic

**Table 1. Preliminary pilot search strategy in MEDLINE/PubMed.**

| MEDLINE/ PubMed | Search strategy |
|---|---|
| | *(P)—Population* |
| | **#1** (("Infant" [MeSH Terms] OR "Infants" [Title/Abstract] OR "Child, Preschool" [MeSH Terms] OR "Preschool Child" [Title/Abstract] OR "Children, Preschool" [Title/Abstract] OR "Preschool Children" [Title/Abstract] OR "Child" [MeSH Terms] OR "Children" OR "Adolescent" [MeSH Terms] OR "Adolescents" [Title/Abstract] OR "Adolescence" [Title/Abstract] OR "Teens" [Title/Abstract] OR "Teen" [Title/Abstract] OR "Teenagers" [Title/Abstract] OR "Teenager" [Title/Abstract] OR "Youth" [Title/Abstract] OR "Youths" [Title/Abstract])) |
| | *(C)—Concept* |
| | **#2** (("Telemedicine" [MeSH Terms] OR "Mobile Health" [Title/Abstract] OR "Health, Mobile" [Title/Abstract] OR "mHealth" [Title/Abstract] OR "Telehealth" [Title/Abstract] OR "eHealth" [Title/Abstract] OR "Telehealth" [Title/Abstract] OR "Teleconsultation" [Title/Abstract] OR "Telemonitoring" [Title/Abstract] OR "Telenursing" [Title/Abstract])) |
| | **#3** (("Chronic Disease" [MeSH Terms] OR "Chronic Diseases" [Title/Abstract] OR "Disease, Chronic" [Title/Abstract] OR "Chronic Illness" [Title/Abstract] OR "Chronic Illnesses [Title/Abstract] OR "Illness, Chronic" [Title/Abstract] OR "Chronic Condition" [Title/Abstract] OR "Chronic Conditions" [Title/Abstract] OR "Condition, Chronic" [Title/Abstract] OR "Chronically Ill" [Title/Abstract])) |
| | *(C)—Context* |
| | **#4** (("COVID-19" [MeSH Terms] OR "COVID 19" [Title/Abstract] OR "COVID-19 Virus Disease" [Title/Abstract] OR "2019-nCoV Infection" [Title/Abstract] OR "2019 nCoV Infection" [Title/Abstract] OR "Coronavirus Disease-19" [Title/Abstract] OR "Coronavirus Disease 19" [Title/Abstract] OR "2019 Novel Coronavirus Disease" [Title/Abstract] OR "2019 Novel Coronavirus Infection" [Title/Abstract] OR "2019-nCoV Disease" [Title/Abstract] OR "2019 nCoV Disease" [Title/Abstract] OR "Coronavirus Disease 2019" [Title/Abstract] OR "SARS Coronavirus 2 Infection" [Title/Abstract] OR "SARS-CoV-2 Infection" [Title/Abstract] OR "Infection, SARS-CoV-2" [Title/Abstract] OR "SARS CoV 2 Infection" [Title/Abstract] OR "SARS-CoV-2 Infections" [Title/Abstract] OR "COVID-19 Pandemic" [Title/Abstract] OR "COVID 19 Pandemic" [Title/Abstract] OR "COVID-19 Pandemics" [Title/Abstract] OR "Pandemic, COVID-19" [Title/Abstract] OR "SARS-CoV-2" [MeSH Terms] OR "2019 Novel Coronavirus" [Title/Abstract] OR "Coronavirus, 2019 Novel" [Title/Abstract] OR "Novel Coronavirus, 2019" [Title/Abstract] OR "SARS-CoV-2 Virus" [Title/Abstract] OR "SARS CoV 2 Virus" [Title/Abstract] OR "2019-nCoV" [Title/Abstract] OR "COVID-19 Virus" [Title/Abstract] OR "SARS Coronavirus 2" [Title/Abstract] OR "Coronavirus 2, SARS" [Title/Abstract] OR "Severe Acute Respiratory Syndrome Coronavirus" [Title/Abstract] OR "SARS-CoV-2 variants" [Supplementary Concept] OR "SARS-CoV-2 B.1.1.7 variant" [Title/Abstract])) |
| | **#5** (("Outpatients" [MeSH Terms] OR "Outpatient" [Title/Abstract] OR "Out-patients" [Title/Abstract] OR "Out patients" [Title/Abstract] OR "Out-patient" [Title/Abstract] OR "Ambulatory Care" [MeSH Terms] OR "Care, Ambulatory" [Title/Abstract] OR "Ambulatory Care Facilities" [Title/Abstract] OR "Outpatient Care" [Title/Abstract] OR "Care, Outpatient" [Title/Abstract] OR "Health Services, Outpatient" [Title/Abstract] OR "Health Service, Outpatient" [Title/Abstract] OR "Outpatient Health Service" [Title/Abstract] OR "Service, Outpatient Health" [Title/Abstract] OR "Outpatient Health Services" [Title/Abstract] OR "Outpatient Services" [Title/Abstract] OR "Outpatient Service" [Title/Abstract] OR "Service, Outpatient" [Title/Abstract] OR "Services, Outpatient" [Title/Abstract] OR "Services, Outpatient Health" [Title/Abstract] OR "Outpatient Clinics, Hospital" [MeSH Terms] OR "Ambulatory Care Facilities, Hospital" [Title/Abstract] OR "Hospital Outpatient Clinics" [Title/Abstract] OR "Clinic, Hospital Outpatient" [Title/Abstract] OR "Clinics, Hospital Outpatient" [Title/Abstract] OR "Hospital Outpatient Clinic" [Title/Abstract] OR "Outpatient Clinic, Hospital" [Title/Abstract])) |
| | **#4:** #1 AND #2 AND #3 AND #4 AND #5 |

conditions during the COVID-19 pandemic will be included. It is noteworthy that there will be no date or language restriction for the study selection.

- *Exclusion criteria*: studies with adults, even if children and adolescents participate; and studies on acute illnesses that address chronic diseases will be excluded.

After searching for studies, articles will be downloaded to the Endnote Web™ bibliographic software to store, organize, manage all references and identify duplicates. The studies will then be exported to the Rayyan™ application, which assists in the screening and selection of studies. At this stage, the titles and abstracts will be read to carry out an initial assessment of the evidence. Two independent researchers (LKDSC and LCLJ) will search and screen the records by titles and abstract into the Rayyan™ app. After the initial screening, the same two independent researchers (LKDSC and LCLJ) will assess the full text of studies retrieved for inclusion/exclusion, using the Rayyan™ app. A third reviewer (RAGL) will decide on any disagreements as to the selected studies.

## Step 4: Charting the data

Two reviewers/authors (LKDSC and LCLJ) will independently chart the data for each included study based on previously published forms [23, 31–35]. The expected date of completion of this scoping review will be December 2022. Information to be extracted includes a) identification of the study and objectives; b) study design; c) study population and baseline characteristics; d) type of exposure; e) study method; f) recruitment methods; g) times of measurement; h) follow-up; i) outcomes; j) main findings; k) clinical and epidemiological significance; and l) conclusions [23, 31–35].

## Results

### Step 5: Collating, summarizing, and reporting the results

A flowchart diagram will describe the entire study selection process [23]. For data analysis, a thematic evaluation will be done to describe the mapping and telehealth strategies and modalities used in outpatient care for children and adolescents with chronic conditions during the COVID-19 pandemic, with a view to reporting which actions were implemented and their impacts on the continuity of care. The systematic assessment of study quality (risk of bias of the included studies) will not be performed, as the intent of this ScR is to provide the mapping and extension of the literature rather than a critical appraisal of the studies like in systematic reviews [23]. Our findings will be presented through tables, charts, narrative summaries, and will be assessed based on the type of data charted and the outcomes. Additionally, the meaning of these findings will be considered as they relate to the guiding question, the characterization and measurement of the impact of different telehealth modalities used in outpatient care for children and adolescents with chronic conditions during the COVID-19 pandemic, and the implications for practice and further research.

## Discussion

The purpose of this ScR is to map the evidence related to telehealth in outpatient care for children and adolescents with chronic conditions during the COVID-19. In addition, the purpose is to identify which strategies were implemented and their impacts on the continuity of care. Thus, we will undertake an ScR. We will search across multiple electronic databases, additional sources, and grey literature. The ScR method serves to chart and map results and to establish the directions for future research.

It is noteworthy that there are two aspects in which the study within this protocol differs from others. Firstly, the ScR is more suitable for our study theme than a systematic review. Our research question is quite broad and focuses on mapping the extension of the evidence available. Secondly, we combine qualitative and quantitative methods to report our results,

using thematic themes and bibliometric trends to provide efficient guidance and meaningful insights into this field.

The use of new and varied methods to review the evidence and collate and summarize our findings represents the strength of our study. To the best of our knowledge, this will be the first ScR that combines these methods to look specifically at the telehealth modalities used in outpatient care for children and adolescents with chronic conditions during the COVID-19 pandemic.

We expect that our results will be of interest to practitioners, researchers, and anyone concerned with this particular emerging issue in the pandemic context. In addition, the plans of dissemination of this study comprise peer-reviewed publication and conference presentations.

## Supporting information

**S1 Checklist. PRISMA-P 2015 checklist.**
(DOCX)

## Author Contributions

**Conceptualization:** Larissa Karoline Dias da Silva Casemiro, Regina Aparecida Garcia de Lima.

**Data curation:** Luís Carlos Lopes-Júnior, Fabrine Aguilar Jardim, Regina Aparecida Garcia de Lima.

**Formal analysis:** Larissa Karoline Dias da Silva Casemiro, Luís Carlos Lopes-Júnior, Fabrine Aguilar Jardim, Regina Aparecida Garcia de Lima.

**Investigation:** Luís Carlos Lopes-Júnior, Mariane Caetano Sulino.

**Methodology:** Luís Carlos Lopes-Júnior.

**Supervision:** Luís Carlos Lopes-Júnior, Regina Aparecida Garcia de Lima.

**Validation:** Luís Carlos Lopes-Júnior, Regina Aparecida Garcia de Lima.

**Visualization:** Larissa Karoline Dias da Silva Casemiro, Luís Carlos Lopes-Júnior, Mariane Caetano Sulino, Regina Aparecida Garcia de Lima.

**Writing – original draft:** Larissa Karoline Dias da Silva Casemiro, Luís Carlos Lopes-Júnior, Fabrine Aguilar Jardim, Mariane Caetano Sulino, Regina Aparecida Garcia de Lima.

**Writing – review & editing:** Larissa Karoline Dias da Silva Casemiro, Luís Carlos Lopes-Júnior, Mariane Caetano Sulino, Regina Aparecida Garcia de Lima.

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
