## [Decision Letter · Decision Letter 0]

15 Mar 2022

PONE-D-21-25612Telehealth in outpatient care for children and adolescents with chronic conditions during the COVID-19 pandemic: a scoping review protocolPLOS ONE

Dear Dr. REGINA APARECIDA GARCIA LIMA

Thank you for submitting your manuscript to PLOS ONE. After careful consideration, we feel that it has merit but does not fully meet PLOS ONE’s publication criteria as it currently stands. Therefore, we invite you to submit a revised version of the manuscript that addresses the points raised during the review process.

We look forward to receiving your revised manuscript.

Kind regards,

Rizaldy Taslim Pinzon

Academic Editor

PLOS ONE

Journal Requirements:

Reviewers' comments:

Reviewer's Responses to Questions

**Comments to the Author**

1. Does the manuscript provide a valid rationale for the proposed study, with clearly identified and justified research questions?

Reviewer #1: Yes

2. Is the protocol technically sound and planned in a manner that will lead to a meaningful outcome and allow testing the stated hypotheses?

Reviewer #1: Yes

3. Is the methodology feasible and described in sufficient detail to allow the work to be replicable?

Reviewer #1: Yes

4. Have the authors described where all data underlying the findings will be made available when the study is complete?

Reviewer #1: Yes

5. Is the manuscript presented in an intelligible fashion and written in standard English?

Reviewer #1: Yes

6. Review Comments to the Author

You may also provide optional suggestions and comments to authors that they might find helpful in planning their study.

Reviewer #1: The protocol titled Telehealth in outpatient care for children and adolescents with chronic conditions

during the COVID-19 pandemic: a scoping review protocol seems to be good study and authors will try to find out ways to follow up chronic cases through telehealth after searching published literatures . Hence in introduction I have come comments.

Line 104: Spell check of highlighted word

Line 110-112: Can be omitted

7. PLOS authors have the option to publish the peer review history of their article (what does this mean?). If published, this will include your full peer review and any attached files.

Reviewer #1: No

---

## [Author Response · Author response to Decision Letter 0]

17 Mar 2022

Response letter to the Editor and reviewers

Ribeirão Preto, SP, Brazil, March 16th , 2022

Coments of Academic Editor, Rizaldy Taslim Pinzon

PONE-D-21-25612

Telehealth in outpatient care for children and adolescents with chronic conditions during the COVID-19 pandemic: a scoping review protocol

PLOS ONE

Dear Dr. Regina Aparecida Garcia Lima

Thank you for submitting your manuscript to PLOS ONE. After careful consideration, we feel that it has merit but does not fully meet PLOS ONE’s publication criteria as it currently stands. Therefore, we invite you to submit a revised version of the manuscript that addresses the points raised during the review process.

Response: Dear Dr. Rizaldy Taslim Pinzon,

We would like to thank you for the opportunity to review the manuscript after the reviewers' suggestions and recommendations.

All points were addressed and/or clarified in this new version. In addition, we responded item by item to the questions raised by the reviewers in this letter.

-Response: It is not necessary. Thanks!

Journal Requirements:

Response: We have carefully reviewed the reference list and we ensure that it is complete and correct. Also, we have no papers retracted cited in this manuscript.

Reviewers' comments:

Reviewer's Responses to Questions

Comments to the Author

1. Does the manuscript provide a valid rationale for the proposed study, with clearly identified and justified research questions?

Reviewer #1: Yes

2. Is the protocol technically sound and planned in a manner that will lead to a meaningful outcome and allow testing the stated hypotheses?

Reviewer #1: Yes

3. Is the methodology feasible and described in sufficient detail to allow the work to be replicable?

Reviewer #1: Yes

4. Have the authors described where all data underlying the findings will be made available when the study is complete?

Reviewer #1: Yes

5. Is the manuscript presented in an intelligible fashion and written in standard English?

Reviewer #1: Yes

6. Review Comments to the Author

You may also provide optional suggestions and comments to authors that they might find helpful in planning their study.

Reviewer #1: 

The protocol titled Telehealth in outpatient care for children and adolescents with chronic conditionsduring the COVID-19 pandemic: a scoping review protocol seems to be good study and authors will try to find out ways to follow up chronic cases through telehealth after searching published literatures . 

Response: Thank you so much for you positive feedback.

- Hence in introduction I have come comments.

- Line 104: Spell check of highlighted word

Response: Done! Thanks for your careful review.

Line 110-112: Can be omitted

Response: We have omitted the sentence as per suggested. Thanks.

Response: Not applicable

The authors

---

## [Decision Letter · Decision Letter 1]

2 May 2022

PONE-D-21-25612R1Telehealth in outpatient care for children and adolescents with chronic conditions during the COVID-19 pandemic: a scoping review protocolPLOS ONE

Dear Dr. REGINA APARECIDA GARCIA LIMA

Thank you for submitting your manuscript to PLOS ONE. After careful consideration, we feel that it has merit but does not fully meet PLOS ONE’s publication criteria as it currently stands. Therefore, we invite you to submit a revised version of the manuscript that addresses the points raised during the review process.

There are some concern about methodological flaws in your study. Please make the revision based on reviewers suggestion and comments. 

We look forward to receiving your revised manuscript.

Kind regards,

Rizaldy Taslim Pinzon

Academic Editor

PLOS ONE

Journal Requirements:

Reviewers' comments:

Reviewer's Responses to Questions

**Comments to the Author**

1. Does the manuscript provide a valid rationale for the proposed study, with clearly identified and justified research questions?

Reviewer #2: Yes

2. Is the protocol technically sound and planned in a manner that will lead to a meaningful outcome and allow testing the stated hypotheses?

Reviewer #2: Yes

3. Is the methodology feasible and described in sufficient detail to allow the work to be replicable?

Reviewer #2: No

4. Have the authors described where all data underlying the findings will be made available when the study is complete?

Reviewer #2: Yes

5. Is the manuscript presented in an intelligible fashion and written in standard English?

Reviewer #2: Yes

6. Review Comments to the Author

You may also provide optional suggestions and comments to authors that they might find helpful in planning their study.

Reviewer #2: In the context of the COVID-19 pandemic, this is certainly an interesting project and well targeted. The title is well aligned with the objectives. However, the methods need an extensive revision. A librarian support is needed in order to render the search strategy more robust.

Introduction:

P3 L110 What is clinical weaknesses? Details or more clarifications are needed.

P3 L130 -P4 L131-133What this is an added value? Use in what context???

Methods

P4, L151-155 The reference of Tricco et al 2018 suffices.

P4 L163-4 what is the added value of this classification? In case it is worth, please elaborate.

P5 L180-1 the definition of outpatient care is very simplistic and may limit the scope of your findings.

P5 L194 certainly assumed. So, no need to mention this “using the Boolean terms AND/OR”.

Table 1 based on the study title and the objectives; some essential key concepts are missing. They are outpatient care and chronic conditions. They are totally ignored.

Table 1 Could authors explain why [All fields] is used. Because it is classical to capture references through titles and abstract. Otherwise, this will lead to incredible volume of « noise ». Please elaborate.

Table 1 Do authors sought a support from librarians? This expertise is essential. #1 AND #2 AND #3 needs AND outpatient…..AND chronic….

The type of design to be included or excluded is not stated.

P5 L222 is it a typo «2021».

P5, L245-6 Please remove this as there is no added value.

P8 L259 In what this is a limitation since you don’t what will be found.

7. PLOS authors have the option to publish the peer review history of their article (what does this mean?). If published, this will include your full peer review and any attached files.

Reviewer #2: No

---

## [Author Response · Author response to Decision Letter 1]

7 May 2022

Response to Reviewers

Ribeirão Preto, SP, Brazil, May 5th, 2022

Minor revision

PONE-D-21-25612R1

Telehealth in outpatient care for children and adolescents with chronic conditions during the COVID-19 pandemic: a scoping review protocol

Dear Dr. Regina Aparecida Garcia Lima

Thank you for submitting your manuscript to PLOS ONE. After careful consideration, we feel that it has merit but does not fully meet PLOS ONE’s publication criteria as it currently stands. Therefore, we invite you to submit a revised version of the manuscript that addresses the points raised during the review process.

There are some concern about methodological flaws in your study. Please make the revision based on reviewers suggestion and comments. 

Response: Dear Dr. Rizaldy Taslim Pinzon,

Thank you for the opportunity to review the manuscript after the reviewers' suggestions and recommendations.

All points were addressed and/or clarified in this new version. In addition, we responded item by item to the questions raised by the reviewers in this letter.

Response: We have carefully reviewed the reference list and we ensure that it is complete and correct. Also, we have no papers retracted cited in this manuscript.

Reviewers' comments:

Reviewer's Responses to Questions

Comments to the Author

1. Does the manuscript provide a valid rationale for the proposed study, with clearly identified and justified research questions?

Reviewer #2: Yes

2. Is the protocol technically sound and planned in a manner that will lead to a meaningful outcome and allow testing the stated hypotheses?

Reviewer #2: Yes

3. Is the methodology feasible and described in sufficient detail to allow the work to be replicable?

Reviewer #2: No

4. Have the authors described where all data underlying the findings will be made available when the study is complete?

Reviewer #2: Yes

5. Is the manuscript presented in an intelligible fashion and written in standard English?

Reviewer #2: Yes

6. Review Comments to the Author

You may also provide optional suggestions and comments to authors that they might find helpful in planning their study.

Reviewer #2: 

In the context of the COVID-19 pandemic, this is certainly an interesting project and well targeted. The title is well aligned with the objectives. 

Response: Thank you so much for your comments.

However, the methods need an extensive revision. A librarian support is needed in order to render the search strategy more robust.

Response: A librarian was consulted for checking the search strategy. We added in this version the complete strategy revised by the librarian of the Federal University of Espírito Santo, respecting the controlled descriptors (subject indexers in each database) and also combining with the synonyms and keywords in order to make the search strategy broad and robust.

Introduction:

P3 L110 What is clinical weaknesses? Details or more clarifications are needed.

Response: We edited this sentence to make it clearer. “Children and adolescents with chronic conditions demand high hospital readmission rates,……”

P3 L130 -P4 L131-133What this is an added value? Use in what context???

Response: We edited this sentence to make it clearer. “Telehealth is being used in several countries worldwide [15–21], especially in non-emergency/routine care and in cases where services do not require direct patient-provider interaction such as providing psychological services, mainly in the current scenario of COVID-19 pandemic in order to nullify the barriers imposed by the COVID-19 pandemic. Additionally, telehealth can become a basic need for the general population, health care providers, and patients with COVID-19, mainly when people are in quarantine, enabling patients in real time through contact with health care provider for advice on their health problems”.

Methods

P4, L151-155 The reference of Tricco et al 2018 suffices.

Response: OK. Done. Thanks!

P4 L163-4 what is the added value of this classification? In case it is worth, please elaborate.

Response: OK. We removed the classification.

P5 L180-1 the definition of outpatient care is very simplistic and may limit the scope of your findings.

Response: OK. Rewriten.

P5 L194 certainly assumed. So, no need to mention this “using the Boolean terms AND/OR”.

Response: OK. Done.

Table 1 based on the study title and the objectives; some essential key concepts are missing. They are outpatient care and chronic conditions. They are totally ignored.

Response: OK. We have added to the search strategy. Many Thanks!

Table 1 Could authors explain why [All fields] is used. Because it is classical to capture references through titles and abstract. Otherwise, this will lead to incredible volume of « noise ». Please elaborate.

Response: OK. We reworded it with the help of the librarian.

Table 1 Do authors sought a support from librarians? This expertise is essential. #1 AND #2 AND #3 needs AND outpatient…..AND chronic….

Response: OK. We reworded it with the help of the librarian.

The type of design to be included or excluded is not stated.

Response: We have added “study design”. Thanks.

P5 L222 is it a typo «2021».

Response: Sure. Thank you. Edited for “2022”.

P5, L245-6 Please remove this as there is no added value.

Response: Ok. Removed.

P8 L259 In what this is a limitation since you don’t what will be found.

Response: Ok. Removed.

The authors

---

## [Editor Report · Decision Letter 2]

31 May 2022

Telehealth in outpatient care for children and adolescents with chronic conditions during the COVID-19 pandemic: a scoping review protocol

PONE-D-21-25612R2

Dear Dr. Lima

We’re pleased to inform you that your manuscript has been judged scientifically suitable for publication and will be formally accepted for publication once it meets all outstanding technical requirements.

Kind regards,

Rizaldy Taslim Pinzon

Academic Editor

PLOS ONE
---

## [Editor Report · Acceptance letter]

2 Jun 2022

PONE-D-21-25612R2 

Telehealth in outpatient care for children and adolescents with chronic conditions during the COVID-19 pandemic: a scoping review protocol 

Dear Dr. de Lima:

I'm pleased to inform you that your manuscript has been deemed suitable for publication in PLOS ONE. Congratulations! Your manuscript is now with our production department. 

Kind regards, 

on behalf of

Dr. Rizaldy Taslim Pinzon 

Academic Editor

PLOS ONE